# *In Medio Stat Virtus*: Moderate Cognitive Flexibility as a Key to Affective Flexibility Responses in Long-Term HRV

**DOI:** 10.3390/s24248047

**Published:** 2024-12-17

**Authors:** Francesca Borghesi, Gloria Simoncini, Alice Chirico, Pietro Cipresso

**Affiliations:** 1Department of Psychology, University of Turin, 10124 Turin, Italy; gloria.simoncini@unito.it (G.S.); pietro.cipresso@unito.it (P.C.); 2Department of Psychology, Research Center in Communication Psychology, Università Cattolica del Sacro Cuore, 20123 Milan, Italy; alice.chirico@unicatt.it

**Keywords:** psychometrics, heart rate variability, *affective storm*, cognitive flexibility, affective flexibility, mental flexibility

## Abstract

This study examines the relationship between cognitive and affective flexibility, two critical aspects of adaptability. Cognitive flexibility involves switching between activities as rules change, assessed through task-switching or neuropsychological tests and questionnaires. Affective flexibility, meanwhile, refers to shifting between emotional and non-emotional tasks or states. Although similar conceptually, prior research shows inconsistent findings on their link, especially regarding physiological and self-reported measures. Affective flexibility was operationalized as changes in heart rate variability (HRV) in response to transitioning affective stimuli, while cognitive flexibility was assessed using self-report questionnaires that captured individuals’ perceived adaptability. The findings revealed that individuals with extremely high or low cognitive flexibility displayed similar HRV patterns in response to emotional stimuli, while those with medium levels of cognitive flexibility showed distinct responses. The Extreme group exhibited higher baseline autonomic activation that decreased after exposure, whereas the Average group’s moderate baseline activation increased post-stimuli. This interaction was mainly seen in long-term HRV indexes, while short-term indexes showed a uniform response across groups, suggesting that differences in flexibility were probably not discernible via state features but rather as traits and long-term attributes. The findings suggest that cognitive and affective flexibility exist on a continuum; extremely high and low cognitive flexibility is linked to intense affective responses, while moderate cognitive flexibility corresponds to balanced physiological regulation, supporting the notion that “*in medio stat virtus*” (virtue lies in moderation).

## 1. Introduction

Flexibility is a key component of human adaptability, allowing individuals to cope with changing environments by dynamically adjusting their thoughts, emotions, and behaviors [1,2]. It encompasses both cognitive and affective processes, and despite its dual nature, flexibility has been interpreted as a continuum of variability, with different components collectively contributing to overall adaptive functioning [3,4,5,6]. Comprehending the interaction of these processes is essential for progressing theoretical frameworks in psychology, since it may uncover common or unique mechanisms that underlie individual adaptation. According to the DSM-5, both high levels of flexibility—often manifesting as impulsivity—and low levels—manifesting as rigidity—can be dysfunctional, as seen in various clinical conditions such as attention-deficit/hyperactivity disorder (ADHD), autism spectrum disorders, and eating disorders [7,8,9]. This relationship between flexibility and functionality follows an inverted U-shaped curve, where optimal functioning occurs at moderate levels of flexibility. At one extreme, excessive flexibility manifests as impulsivity and difficulty maintaining stable patterns of behavior, as seen in ADHD, where individuals struggle to inhibit responses and maintain consistent routines [7]. This heightened flexibility can lead to difficulties in maintaining focus, completing tasks, and establishing stable behavioral patterns. At the other extreme, insufficient flexibility presents cognitive and behavioral rigidity, characteristic of autism spectrum disorders, where individuals may struggle to adapt to changes in routine or shift between different perspectives [8]. In eating disorders, both extremes can be observed, namely a rigid adherence to strict dietary rules and inflexible thought patterns about body image, or impulsive eating behaviors and difficulty maintaining consistent dietary patterns [10,11]. The optimal range of flexibility lies between these extremes, allowing individuals to maintain stable patterns while adapting appropriately to environmental demands. Both behavioral and physiological evidence suggest that individuals at the extremes of flexibility—both cognitive and affective—tend to show similar patterns. Specifically, individuals with either extreme flexibility—very high or very low—tend to exhibit reduced adaptability and increased physiological reactivity (e.g., lower heart rate variability), suggesting shared limitations in their capacity to regulate responses effectively [12,13,14,15,16] (Figure 1). This evidence possesses considerable clinical significance, since deficiencies in cognitive and affective flexibility are characteristic features of different psychopathological conditions. Examining their interconnection may enhance diagnosis accuracy and guide interventions designed to promote adaptation in these domains.

In this complex landscape, studying the relationship between affective flexibility and cognitive flexibility poses challenges. Cognitive flexibility (CF) is a multifaceted construct including two main domains, one associated with executive functioning and the other with relational and communicative abilities. Both involve shifting between different task sets in response to changing rules. CF is assessed through neuropsychological tests, such as the Trail Making Test (TMT) [17], the Wisconsin Card Sorting Test (WCST) [18], and the Stroop Test [19], which evaluate attentional and executive processes. Additionally, self-report questionnaires like the Cognitive Flexibility Inventory (CFI) [20], the Cognitive Flexibility Scale (CFS) [21], and the Cognitive Control and Flexibility Questionnaire (CCFQ) [22] provide complementary insights into individuals’ perceptions of their adaptability in various contexts.

In the executive functioning framework, affective flexibility (AF) is often defined as an aspect of cognitive flexibility, focusing on the ability to alternate between emotional and non-emotional task sets based on situational demands. Conversely, the emotion regulation framework defines AF as the variability in one’s emotional states or affective dynamics [23,24,25]. The executive functioning approach conventionally measures AF through task-switching paradigms using emotional datasets like the International Affective Picture System (IAPS) or FACES [26], with switching costs reflecting flexibility. Lower switching costs signify greater affective flexibility. From the emotion regulation perspective, AF is assessed in terms of the ability to shift between different emotional states using laboratory measures or experience sampling methods (ESMs) [27,28]. While laboratory settings measure the effects of pre- and post-stimuli exposure on affective states, ESMs capture moment-to-moment affective changes in real life, though without direct links to environmental stimuli.

Studies have primarily investigated the relationship between cognitive and affective flexibility within the executive function framework using task-switching measures that produced inconsistent results. For instance, research has shown that more efficient shifting between non-affective and affective information is linked to better emotional regulation [29,30], while others reported no clear correlation between cognitive and affective flexibility, or only minor associations [31,32]. Thus, no clear evidence exists as to whether cognitive and affective flexibility represent similar or distinct processes.

There is a clear lack of studies exploring the relationship between self-reported cognitive and affective flexibility. One reason is that existing cognitive flexibility measures (e.g., CFI, CFS, CCFQ) already contain items that touch upon emotional aspects (see Table 1). For example, the Cognitive Flexibility Inventory (CFI), a 19-item measure divided into “Alternatives” and “Control” subscales, includes items such as “I try to think about things from another person’s point of view”, such as “I am good at sizing up situations”, which reflect an individual’s ability to take on others’ perspectives and handle emotional changes—crucial components of empathy and emotional adaptability. Similarly, the Cognitive Flexibility Scale (CFS), a 12-item measure focused on interpersonal communication competence, contains statements like “I am willing to listen and consider alternatives for handling a problem”. These statements imply emotional adaptation in challenging situations, reflecting one’s emotional resilience and capacity to adapt when facing obstacles. The Cognitive Control and Flexibility Questionnaire (CCFQ) an 18-item measure, evaluates both cognitive control over emotions and appraisal/coping flexibility. It includes items such as “Generally, in stressful situations I can easily think of multiple coping options before deciding how to respond”. These items highlight adaptability in emotional situations, showing how well an individual can cope with unforeseen challenges potentially inducing stress or anxiety. Such abilities explicitly relate to emotional regulation and reflect the capacity to manage emotional responses in stressful situations—a key component of affective flexibility.

Moreover, correlating self-reported cognitive flexibility with reaction time-based measures of affective flexibility (e.g., switching costs) has yielded non-significant results [33]. The inconsistencies emphasized the need for innovative and integrated methodologies that go beyond the separation of cognitive and affective components.

Hence, this study aims to investigate the relationship between self-reported cognitive flexibility and a new measure of affective flexibility using heart rate variability (HRV). Our approach adopts Hollenstein’s [23] and Koval’s [25] definition of affective flexibility as the ability to transition between different affective states. We developed an experimental design in a laboratory setting to recreate these transitions, capturing their physiological manifestation through cardiac variability.

The idea is to expose participants to an *affective storm*—a sequence of emotion-eliciting stimuli from the International Affective Picture System (IAPS) designed to replicate the transition between different affective states in a controlled laboratory setting. This approach diverges from conventional stress-based [34] paradigms by focusing on non-stressful affective stimulation. Our objective is to capture the dynamic and fluid nature of emotional transitions rather than acute stress responses. The *affective storm* consists of carefully selected positive and negative stimuli that evoke various affective states.

We propose measuring HRV before and after exposure to this *affective storm* as an index of affective flexibility. HRV refers to the variation in time intervals between heartbeats, reflecting the parasympathetic (vagal) branch of the autonomic nervous system. The vagal system’s rapid responsiveness allows for moment-to-moment modulation of cardiac activity, especially in reaction to emotional stimuli [35]. While resting HRV indicates baseline autonomic regulation, changes in HRV in response to emotional events provide additional insights, highlighting the body’s adaptive capacity to shift physiological states according to situational demands. We define an adaptive HRV response as an increase during the post-exposure phase after the cessation of emotional stimulation. Conversely, a consistently low or blunted HRV response indicates poorer adaptability [36,37].

Several studies have previously investigated HRV in relation to cognitive and affective flexibility. Past research has shown a consistent link between autonomic regulation and cognitive flexibility, with higher levels of resting HRV associated with enhanced cognitive functions such as improved monitoring and updating of working memory, enhanced attention control, and greater response inhibition [38,39,40,41].

Grol and De Raedt [13,34] investigated the relationship between affective flexibility (defined in the executive function framework and measured with an affective switching task) and HRV, but their findings were inconclusive. In their 2020 study [13], the authors unexpectedly found an inverse relationship, where a higher resting HRV was negatively associated with more efficient shifting away from negative affective information (greater affective flexibility). This counterintuitive finding suggested that individuals exhibiting more flexibility in avoiding unpleasant stimuli had lower resting HRV, possibly due to strategic emotional avoidance. In their 2021 study [34], Grol and De Raedt expanded the analysis to examine HRV during stress induction and recovery phases. However, they found no significant association between affective flexibility and HRV responses during or after stress, indicating that affective flexibility did not directly impact physiological regulation in these contexts.

In light of these divergent results, we decided to implement a different measuring paradigm that focuses on transitions between various affective states rather than the affective task-switching paradigm. In line with Hollenstein’s [23] definition of affective flexibility, our experimental design involves an *affective storm* to recreate emotional transitions and measure HRV before and after exposure. By adopting this non-stressful affective stimulation approach, we aim to assess the dynamic nature of affective transitions and investigate the relationship between HRV and self-reported cognitive flexibility using validated questionnaires, as the CFI, CCFQ, and CFS.

The evidence suggests that flexibility operates along a continuum [12,42], where both extremes—rigidity and excessive flexibility—may be dysfunctional. Consequently, we hypothesize that individuals with the highest and lowest levels of cognitive flexibility exhibit similar psychophysiological patterns, while those with medium levels will demonstrate greater affective adaptability. Participants were divided into two main groups based on their ranking cognitive flexibility scores, namely Extreme (individuals with either the highest and lowest levels) and Average (individuals with mid-range levels). We expected this grouping to reveal complex dynamics between cognitive and affective flexibility, potentially highlighting shared characteristics across extremes and distinct traits at medium levels.

This study aims to contribute to a more nuanced understanding of cognitive–affective flexibility by exploring both dimensions in tandem rather than considering them as isolated constructs. We also seek to provide a clearer picture of the continuum along which flexibility operates, with the ultimate goal of identifying adaptive patterns that contribute to optimal functioning.

## 2. Materials and Methods

### 2.1. Participants

A cohort of 44 (25 women) adults voluntarily took part in the experiment. The men’s mean age was 26 (SD = 3.59), while the women’s mean age was 24.77 (SD = 5.80). The study by Grol and colleagues [13] was used to estimate the effect dimension. The minimum sample size required to test the study hypothesis was determined using G*Power version 3.1.9.7 [43] through a priori power analysis. Results indicated that the required sample size to achieve 95% power for detecting a medium effect size of 0.30, at a significance criterion of α = 0.05, was N = 40 for an ANOVA mixed with within–between interaction. In light of dropouts and recording difficulties pertaining to physiological signals, we enlisted a total of 44 subjects.

The study was carried out in adherence with the Declaration of Helsinki after receiving approval from the Ethics Committee of the University of Turin (prot. no. 0657478).

### 2.2. Inclusion Criteria

All subjects provided written informed permission prior to participation in the trial. The study’s inclusion criteria required participants to be at least 18 years old and report no psychiatric disorders. Furthermore, participants must be free of cardiovascular or neurological disorders or cardiac arrhythmias.

### 2.3. Procedure

The experimental method was divided into a cognitive phase and an affective phase. The cognitive phase involved self-report questionnaires assessing cognitive flexibility, while the affective phase focused on exploring the physiological activation of affective flexibility, such as HRV.

Throughout the cognitive phase, the following self-reports were administered to test cognitive flexibility [44,45,46]:-Cognitive Flexibility Inventory—CFI [20,47]: This consists of 19 items that form two different subclasses, which are Alternatives (12 items) and Control (7 items), which are rated on a 7-point Likert scale (1 = strongly disagree, 7 = strongly agree). The “Alternatives” sub-dimension assesses the capacity to produce several answers to issues and to consider circumstances from different viewpoints. On the other hand, the “Control” sub-dimension evaluates the extent to which an individual believes in their capability to successfully implement these alternative methods in various situations. In general, higher scores on the overall measure represent high cognitive flexibility.-Cognitive Control and Flexibility Questionnaire—CCFQ [22]: The 18-item questionnaire measures an individual’s perceived ability to exercise control over intrusive, unwanted (negative) thoughts and emotions (sub-component named as “Cognitive Control over Emotion”) and their ability to cope flexibly with a stressful situation (sub-component named as “Appraisal and Coping Flexibility”). The rating for each item is measured using a 7-point Likert scale, where 1 represents severe disagreement and 7 represents strong agreement.-Cognitive Flexibility Scale—CFS [21]: This consists of 12 items and was developed to measure the components of cognitive flexibility related to interpersonal communication competence. Each item is scored using a 6-point Likert scale (1 = strongly disagree, 6 = strongly agree). The CFS does not consist of separate sub-dimensions but rather assesses the overall quality of adaptability in communication. Interpersonal communication flexibility refers to an individual’s capacity to adapt their communication style based on the social situation and the requirements of the encounter, indicating a wide range of flexibility in interpersonal interactions.

These instruments, examining different facets and aspects of cognitive flexibility, enabled us to consider cognitive predispositions that could potentially impact how individuals perceive and react to emotional inputs.

For the affective phase, we adopted Hollenstein’s [23] definition of affective flexibility, which describes it as the capacity to transition smoothly between different emotional states. Our goal was to recreate this concept in a laboratory setting, which led us to utilize Russell’s circumplex model of affect as the foundation for our experimental design. In fact, based on Russell’s model, there are twelve possible transitions between different combinations of arousal and valence levels [48,49,50]. According to this model, four quadrants are defined by the intersection of arousal and valence levels; Quadrant “A” represents “Stress” (high arousal and negative valence), Quadrant “B” represents “Engagement” (high arousal and positive valence), Quadrant “C” represents “Boredom” (low arousal and negative valence), and Quadrant “D” represents “Relaxation” (low arousal and positive valence). Transitions between these quadrants can occur horizontally, vertically, or diagonally. Throughout the session, the participant observed a series of emotionally evocative images, divided into blocks, each representing quadrants of Russel’s circumplex model. The 13 blocks for each subject sequence reflect the total block useful to mimic all the 12 possible transitions between quadrants (Figure 2). Each block contains 12 images, each lasting 10 s, for a total of 156 images and 1560 s of experiment duration. The images were taken from the IAPS, a picture collection specifically designed to evoke emotional reactions [26]. The original images are classified as pleasant, neutral, or unpleasant, and used to statically elicit affect states. Each IAPS image had assigned standardized values for arousal and valence, using the 9-point Likert scale of the Self-Assessment Manikin (SAM) [51]. We meticulously selected images that have the following characteristics: a notable degree of arousal/valence for the condition of high arousal and positive valence, as evidenced by Likert point ratings above 6, and a minimal degree of arousal/valence, as indicated by Likert point ratings below 4, for the condition of negative valence or low arousal. Based on this criterion, 50 images were chosen for quadrant A, 48 images for quadrant B, 46 images for quadrant C, and 149 images for quadrant D. They were then randomly placed into the 13 blocks.

The goal was to expose individuals to various transitions between emotional states, creating what could be termed an *affective storm*. This task, previously developed [52] and applied to an experimental study [53], relies on the idea that affective flexibility can be examined through the lens of affective dynamics, reflecting the notions posited by Hollenstein [23] and Koval [25], who conceptualized affective flexibility as the capacity to shift between different affective states. By inducing an *affective storm*, we aimed to explore the spectrum of emotional states and their dynamic transitions within this model. Our experimental setup involves simulating and analyzing all 12 possible transitions between the four quadrants, effectively inducing controlled emotional shifts in participants and examining their ability to navigate these shifts in a laboratory environment.

The image block sequence for each participant was randomized to minimize any possible order effects. There was a two-minute baseline phase before (T0) and after (T1) in the experiment to check the subjects’ trait (T0) and state (T1) physiological responses. During the resting state phases, participants were presented with a standardized visual stimulus consisting of a white fixation cross displayed centrally on a black background via a computer monitor. Subjects were instructed to maintain their gaze on the fixation point while engaging in spontaneous respiration, with explicit directions to refrain from verbal communication or movements to minimize artifacts. We are specifically interested in the pre- and post-trial baseline in order to test whether there are any differences between different levels of cognitive flexibility (Extreme vs. Average) after being subjected to different emotional transitions, so a kind of *affective storm*.

This study used blood volume pulse (BVP) signals as a metric for assessing *affective flexibility* responses to emotional expressions.

### 2.4. Recording of Psychophysiological Signals

The data related to the activity of the autonomic nervous system were collected by measuring a physiological response of blood volume pulse (BVP). Nexus-10 acquired these responses. The responses were then processed with customized software developed using MATLAB 9.13.0 (R2022b), [54]. Every channel was acquired synchronously at 1024 Hz and extracted at 1024 Hz for the computation of indexes.

### 2.5. Psychophysiological Signal Processing

In a continuous BVP record, each peak-to-peak complex is detected and visually inspected to correct missing data by interpolation. The study meticulously inspected the signal quality of physiological data on a subject-by-subject basis to ensure its integrity. All subjects, except one, displayed clear, interpretable signals without needing beat correction or noise removal. The measuring device’s ineffective signal acquisition compromised the data of one participant, who had their data marked as missing. The signal processing involved a 50 Hz BVP notch filter to eliminate power line interference and the Pan–Tompkins algorithm for detecting IBI complexes.

We analyzed the inter-beat interval (IBI) derived from the blood volume pulse sensor, which corresponds to the interval between RR peaks in the ECG. The inter-beat interval (IBI), also known as RR, was transformed into an estimation of heart rate (HR) and pulse amplitude (BVP amplitude), which reflect the proportional rise in blood volume. We took out the RR time series, interpolated it at a rate of 4 Hz, and got rid of any non-linear trends by smoothing the RR intervals and detrending them with a smooth parameter of 500. These methodological steps ensured the quality and reliability of the physiological data for subsequent analyses. The HR data of BVP were denoted as HR mean (beats per minute) and RR mean (60,000/HR). To evaluate the reaction of the autonomic nervous system, the Task Force of the European Society of Cardiology and the North American Society of Pacing and Electrophysiology suggests the extraction of standard temporal, spectral, and non-linear HRV indexes [55,56].

HRV time domain indexes measure the amount of change in inter-beat interval (IBI) measures, which show the time between heartbeats. As a measure in the temporal domain, we computed the root mean square of the successive differences (RMSSD), the standard deviation of NN intervals (SDNN), and the standard deviation of the heart rate (SD HR) [3]. These indexes measure variability over a period, indicating the autonomic nervous system’s (including both sympathetic and parasympathetic influences) ability to respond to different stimuli. Specifically, the RMSSD is a robust measure of parasympathetic (vagal) activity, reflecting the influence of the autonomic nervous system on HR. RMSSD is widely used in both clinical settings and research to monitor heart health, assess stress levels, and evaluate the impact of interventions on autonomic function [13]. SDNN reflects both long-term and short-term variations in HR and it is associated with both general emotional states and the long-term mood system [57]. Given these characteristics, these indexes seemed useful for our research. These measures collectively show how the HR varies over time, indicating the body’s flexibility in responding to different situations [56].

Frequency domain measurements assess the distribution of absolute or relative power across three frequency bands. We used the LF/HF ratio as an indicator of general autonomic modulation [58]. This index implicates interpretative challenges, since it shows limited capacity to fully capture autonomic dynamism in terms of the balance between the sympathetic and parasympathetic system [41,59,60,61].

The non-linear domain enables the measurement of the unpredictability of a time series by graphing each RR interval on a Poincaré plot. Poincaré plot analysis allows researchers to visually examine concealed patterns within time series, which are sequences of data obtained from sequential measurements. Poincaré plot analysis is insensitive to RR interval trends, unlike frequency domain observations. We estimated that the standard deviation (SD) of the distance between each point and the y = x axis (SD1) determines the width of the ellipse. Additionally, the standard deviation of each point from the y = x + average RR interval (SD2) determines the length of the ellipse. Referring to these two indexes, we focused on the ratio of SD2/SD1, which quantifies the variability of the RR time series used to assess autonomic balance [62,63,64,65]. This non-linear measure provides additional insights into the complexity and variability of HR dynamics. It is useful for exploring how emotions influence the predictability and structure of HRV, offering a more nuanced understanding of autonomic regulation in relation to different affective states [57,63].

### 2.6. Statistical Analyses

Analyses were performed using Jamovi Statistics software (version 2.3.21). Two normality tests (i.e., Kolmogorov–Smirnov and Shapiro–Wilk) were performed, determining a normal distribution of variables related to cognitive and affective flexibility. Conditions (Extreme vs. Average) were calculated with a ranking analysis, using all cognitive flexibility scales, e.g., the alternative dimension of the CFI, control dimension of the CFI and CFS, control over emotion of the CCFQ, and the alternative view of the CCFQ. Within our analytical framework, each participant was given four separate scores for cognitive flexibility, which align with the previously indicated sub-dimensions across several assessments. We computed rankings for each of the four flexibility scores and then categorized subjects into three subgroups, namely “High Flex” (who correlated above the 75th), “Average” (afferents at the 50th percentile), and “Low Flex” (below the 25th percentile). The segmentation was determined by evaluating their performance in individual domains of cognitive flexibility. Each subject was categorized based on their rankings across the four tests, ensuring that they exhibited consistent levels of cognitive flexibility in at least three of the four measures. By employing this method, we were able to categorize each participant into one of three groups—High Flex, Average, or Low Flex—with a significant degree of confidence in the reliability of their cognitive flexibility profile. Our prior hypothesis, also supported by the results of the literature, is how extremes (high and low) can have opposite characteristics from a behavioral point of view (either rigidity or extreme impulsiveness) but are physiologically similar in contrast to the averages.

Afterward, in order to confirm our prior hypothesis, we investigated whether individuals classified as “Low” (below the 25th percentile) and “High” (above the 75th) in terms of cognitive flexibility displayed comparable psychophysiological patterns using a Bayesian *t*-test in the pre-experimental condition. The Bayesian t-test results showed that the evidence was greater than 2 for all physiological indicators of affective flexibility. This is strong evidence supporting the concept that the physiological reactions associated with affective flexibility are comparable among individuals with significant differences in cognitive flexibility (high vs. low) (see Appendix A); subjects with opposite characteristics in terms of cognitive flexibility (high vs. low) seems to have the same physiological pattern in terms of affective flexibility, reasonably demonstrating the choice of joining the extremes (high and low) in a unique group vs. average cognitive flexibility.

## 3. Results

We used traditional null hypothesis significance testing (NHST) [66] to test the difference in the flexibility level (Extreme vs. Average) in terms of changes between pre- and post-*affective storm* induced by images’ emotional transition visualization. In Table 2 and Table 3, we report descriptive cognitive and affective flexibility variables, divided into behavioral and physiological measurements used.

In our study, we analyzed the data using a mixed ANOVA model. This statistical approach was chosen due to its suitability for our experimental design, which involved both between-subject and within-subject variables. The between-subjects factor was the participants’ grouping based on their response to the cognitive flexibility scales, categorizing them into two groups, namely “Average” (median performers at the 50th percentile) and “Extreme” (high performers above the 75th percentile merged with low performers below the 25th percentile). The within-subjects factor was time, divided into “pre-experiment baseline” and “post-experiment baseline” phases, allowing us to examine changes in HRV linked to affective flexibility after exposure to emotional stimulation.

### 3.1. Temporal Emotional Dynamics

In examining the temporal dynamics of emotional response through HRV indexes, a significant main effect of time was observed. Specifically, a notable increase in cardiac variability over time was evident across both groups in the study. This temporal effect was statistically significant for the following variables: SDNN, showing increased variability over time, F(1, 41) = 6.630, *p* = 0.014, $\eta_{p}^{2}$ = 0.139; and SD HR, further supporting the observed trend, F(1, 41) = 6.140, *p* = 0.017, $\eta_{p}^{2}$ = 0.130. As depicted in Figure 3, these findings suggest a shared increase in HRV over time across subjects, indicating a dynamic temporal component in the emotional regulation processes. Although no statistically significant differences emerged in the RMSSD, this index showed the same trend as the others.

### 3.2. Spectral Emotional Dynamics

The interaction effects within the spectral components of heart rate variability, illustrated in Figure 4, revealed significant findings. For the low-frequency/high-frequency ratio (LF/HF ratio), the interaction effect between time and group was also significant, F(1, 41) = 4.92, *p* = 0.032, $\eta_{p}^{2}$ = 0.107.

### 3.3. Non-Linear Emotional Dynamics

In examining non-linear dynamics (Figure 5), the SD2/SD1 ratio showed a significant time * group interaction, F(1, 41) = 4.46, *p* = 0.041, $\eta_{p}^{2}$ = 0.098.

## 4. Discussion

This study investigated the complex connection between affective flexibility, measured through HRV, and cognitive flexibility, as assessed with self-report questionnaires.

The investigation introduces two novel components; the first pertains to the evaluation of affective flexibility via an experimental laboratory design that simulates all potential affective transitions through the induction of an *affective storm*, measuring the resultant changes with cardiac variability, which effectively reflects both short- and long-term physiological alterations associated with emotionality. The second new aspect is the examination of the link between cognitive flexibility, assessed using self-reported questionnaires, and the physiological correlates of affective flexibility.

Ambiguous and intriguing evidence from the work of Grol et al. [13,34] found mixed results in the relationship between affective flexibility (as measured by a task-switching task and reaction time) and indexes of cardiac variability, particularly the RMSSD, a temporal HRV signal similar to SDNN. Our hypothesis asserts that extremely high and low levels of both high and low cognitive flexibility may exhibit similar patterns of physiological activity, since flexibility operates on a continuum [12,42]. This new perspective offers an explanation for Grol’s initially non-intuitive data, suggesting that the extremes at both high and low ends might have a similar distribution of HRV indexes, representative of affective flexibility.

In the temporal indexes, we observed a significant main effect of time, with both Extreme and Average cognitive flexibility groups demonstrating increased cardiac variability over time, particularly notable in short temporal indexes as SDNN (ms) and SD HR measures. SDNN (ms), the standard deviation of all RR intervals (SDNN) reflecting all the cyclic components responsible for variability and the SD HR (the standard deviation of HR) are both used as indicators of short-term variation in HRV. An increase in SDNN and SD HR values suggests an enhancement in HRV in a short time, which indicates the heart’s ability to respond to various physiological and environmental stimuli. A higher HRV is often interpreted as a sign of good cardiovascular health and an efficient autonomic nervous system, which is responsible for the involuntary regulation of body functions, including the stress response [56,67,68]. SDNN represents the standard deviation of all the intervals between consecutive heartbeats (NN intervals). A higher SDNN value indicates greater variation between these intervals, suggesting that the heart can quickly and effectively adapt to various changes such as stress or physical exercise. This is important because a more responsive and flexible heart is better suited to managing stress without overburdening itself. SD HR is similar to SDNN but focuses on the variation in HR rather than the intervals between beats. An increase in SD HR indicates that the HR substantially varies during the monitoring period. This increase could reflect the short-term cardiac physiological immediate response in terms of recovery following exposure to the continuous stimulation of an affective storm. These results align with previous studies on emotions, which have similarly reported heightened activations during and post-task phase under comparable conditions [69,70]. Heightened activations may serve as a generalized adaptive mechanism, reflecting a short-term cardiac physiological response aimed at recovery following the continuous stimulation of an *affective storm*. Such short-term cardiac responses likely represent a broader pattern of recovery dynamics, where the physiological system momentarily amplifies activity to address the challenges presented by extreme emotional stimuli [71,72]. The experimental design, which included highly evocative stimuli varying in emotional intensity, supports this interpretation, as it highlights the role of extreme stimuli in eliciting pronounced, albeit temporary, physiological adjustment.

The same conclusions cannot be made for spectral and non-linear indexes. Our study found interaction effects for the spectral indexes such as the LF/HF ratio and non-linearity such as the SD2/SD1 ratio, which serve as potential markers capable of classifying and distinguishing between Extreme and Average groups. The LF/HF ratio and the SD2/SD1 are key indexes in the evaluation of the autonomic balance, which is the interplay between the sympathetic and parasympathetic branches of the autonomic nervous system. The LF (low frequency) component is often associated with both sympathetic activity and vagal activity, whereas the HF (high frequency) component is typically associated with parasympathetic (or vagal) activity. The SD2/SD1 ratio comes from Poincaré plot analysis, which is a non-linear method to assess HRV. SD1 measures short-term HRV, and it is mainly influenced by parasympathetic activity, while SD2 measures long-term HRV and reflects the autonomic balance. A higher SD2/SD1 ratio suggests a dominance of long-term variability over short-term variability, which emphasizes longitudinal adaptability rather than acute reactivity [64,65].

These measures are significant, as they provide insight into how individuals respond to stressors and emotional stimuli [73]. Emotions are associated with physiological responses that are mediated by the autonomic nervous system. For example, stress-related emotions often increase sympathetic activation, preparing the body for a “fight or flight” response. In contrast, calm or restorative emotions enhance parasympathetic output, promoting a “rest and digest” state. In our study, the observed interaction effects for the spectral LF/HF ratio, as well as the non-linear index SD2/SD1, indicate that exposure to a dynamic range of emotional stimuli result in a significant reduction in these indexes among Extreme individuals [56,74,75]. The comparable (respectively SDNN and SD HR indexes) and contrasting (LF/HF ratio and SD2/SD1) emotional response of Extreme individuals compared to the Average group, especially after the extensive emotional stimulation of an *affective storm*, revealed interesting patterns in autonomic regulation. An intriguing relationship emerged when analyzing parasympathetic activation independently of the ratios, focusing on the individual indexes. At the descriptive level (as shown in Table 3), the Extreme group, while maintaining stable sympathetic activation (LF power and SD2), showed an increase in parasympathetic components (HF power and SD1) after the *affective storm*. This adjustment kept the ratios nearly unchanged between pre- and post-storm measurements. In contrast, the Average group exhibited a greater activation of sympathetic components (LF power and SD2) and a decrease in parasympathetic components post-affective stimulation, leading to an increase in the ratios after the *affective storm*. This distinct psychophysiological behavior between the Average and Extreme group recalls broader principles of human adaptation and functionality. The former pertains to the individual’s adaptive reactions to environmental situations, whilst the latter underscores the significance of these responses in improving the individual’s well-being in a specific context. Thus, adaptability does not inherently reflect a functional response to a particular setting [76,77]. In this context, both physiological behaviors observed in the spectral and non-linear indexes of the two groups can be considered adaptive. The Extreme group’s response pattern suggests a complex interplay between sympathetic stability and enhanced parasympathetic activation, while the Average group showed a more straightforward increase in overall autonomic activation after the *affective storm*. Both patterns appear adaptive, as they reflect appropriate responses to the stimuli and the respective baseline states of the groups. The key question, however, lies in determining which of these responses is also functional. Future research, particularly involving clinical populations, will be critical in exploring this aspect and understanding how these adaptive patterns relate to well-being in specific contexts.

Our results suggest that cognitive and affective flexibility could operate along a continuum, with both extremes displaying similar psychophysiological patterns, thus challenging the traditional dichotomy of these constructs. This continuum perspective provides insight into the nuanced role that cognitive flexibility plays in emotional regulation and adaptation. Our study reinforces the concept that optimal levels of flexibility likely exist within a balanced middle ground, avoiding the dysfunctions associated with rigidity and excessive flexibility. The observed patterns in the Extreme group, when subjected to a wide range of emotional stimuli, suggest a regulatory mechanism that modulates their initial autonomic activation. These findings may have implications for understanding emotional regulation in clinical populations.

## 5. Conclusions

Quantifying and characterizing flexibility, whether behavioral, physiological, or affective, is consistently intricate. How can one capture something that is inherently dynamic?

The research explores flexibility in its dimensions of cognitive–affective properties, involving adaptive variability to respond to environmental changes [68,78,79,80]. It emphasizes the ability to modify and update information and adapt physiological and behavioral reactions to various circumstances, including relationships and emotions. Flexibility is defined by persistent behavioral variety and develops along a continuum [12,42].

Our findings revealed that individuals with both extremely high and extremely low levels of self-reported cognitive flexibility show similar physiological responses in terms of affect flexibility to *affective storms* triggered by emotional images. The main long-term autonomic indexes, namely the LF/HF ratio and SD2/SD1, exhibited opposing physiological activation patterns between the groups; when descriptively analyzing the individual indexes of both ratios, a distinct pattern emerges. In the Extreme group, sympathetic activation (LF power and SD2) remains stable before and after the *affective storm*, while parasympathetic activation increases, resulting in no changes to the LF/HF ratio. In contrast, the Average group exhibits increased sympathetic activation and decreased parasympathetic activation after the *affective storm*. The Extreme group behavior may indicate a tendency to maintain a static autonomic balance regardless of the intensity of the emotional stimulus, which could be advantageous in certain situations but limiting in others. Conversely, the Average group responds to the emotional storm with a significant increase in sympathetic activation and a decrease in parasympathetic activation. This modulation might reflect greater sensitivity to the emotional context and dynamic adaptation, potentially useful for dealing with diverse emotional events.

The study demonstrates that when temporal cardiac variability measures are considered, the interaction effect is abolished and the main effect of time becomes dominant. The short-term indexes exhibited minimal disparity between the extremes and averages, thus clarifying the findings of a prior study conducted by Grol et al. [13]. The immediate emotional reaction between the two groups was typically similar, but the long-term consequences exhibited contrasting activation trends. The subject’s flexibility characteristics seem to be intrinsic rather than transient, influencing long-term behavioral and physiological responses. The research could potentially contribute to a broader understanding of human adaptability, demonstrating that flexibility is not merely a cognitive process but a complex interplay of physiological, emotional, and behavioral mechanisms. Analyzing cardiac variability and emotional reactions may reveal the intricate methods by which people adjust their internal states in reaction to external stimuli.

Future research should investigate the intricate relationship between cognitive and affective flexibility using ecological, behavioral, and direct approaches. For example, 360° videos or virtual reality could be employed to replicate real-life situations in which individuals are required to make decisions [52,81]. Similarly, the use of longitudinal methods of data collection could contribute to a greater understanding of the dynamics between affective and cognitive flexibility over the long term. These approaches would facilitate individualized and patient-centered rehabilitation procedures, specifically for cognitive impairments in executive functions such as Parkinson’s and Alzheimer’s disease, as well as emotional regulation impairments like depression, anxiety, or eating disorders [29,52,82,83]. This would facilitate a more comprehensive understanding of the intricate correlation between cognitive and affective flexibility.

## 6. Limitations

This study has some limitations that should be taken into account. Executed in a controlled laboratory setting, it may not entirely represent the functioning of cognitive and emotional flexibility in real-world circumstances, which are often more intricate and unexpected. Furthermore, given its cross-sectional nature, the present research offers a snapshot of the relationship between cognitive and affective flexibility. Regarding HRV indexes, while we utilized the LF/HF ratio as one of our metrics, it is important to acknowledge ongoing debates in the field regarding its interpretation as a complex index. Future research might incorporate advanced methodologies, such as 2D scatter plots, to address these limitations and provide a more nuanced understanding of autonomic modulation [62]. An additional limitation concerns the 2 min duration of our HRV resting-state recordings. While this timeframe allowed us to minimize participant fatigue during exposure to emotionally evocative stimuli, longer recording periods might have provided more robust HRV data. Future studies may benefit from exploring extended measurement intervals while carefully considering the trade-off between comprehensive physiological assessment and participant engagement.

## Figures and Tables

**Figure 1 sensors-24-08047-f001:**
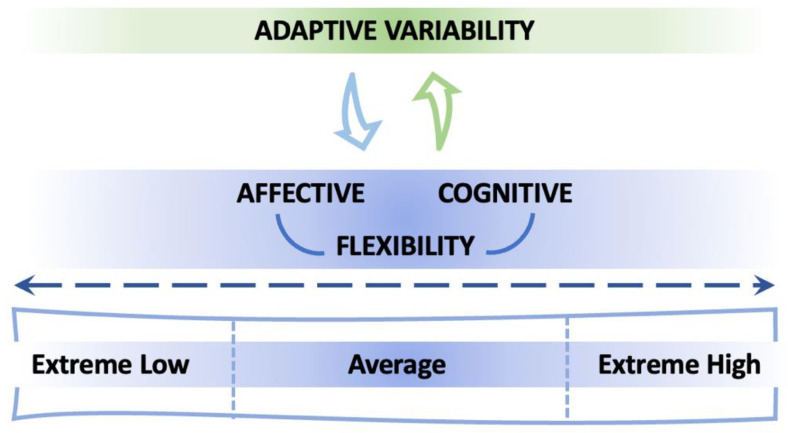
Illustrative representation of the link between adaptive variability and the continuum of flexibility.

**Figure 2 sensors-24-08047-f002:**
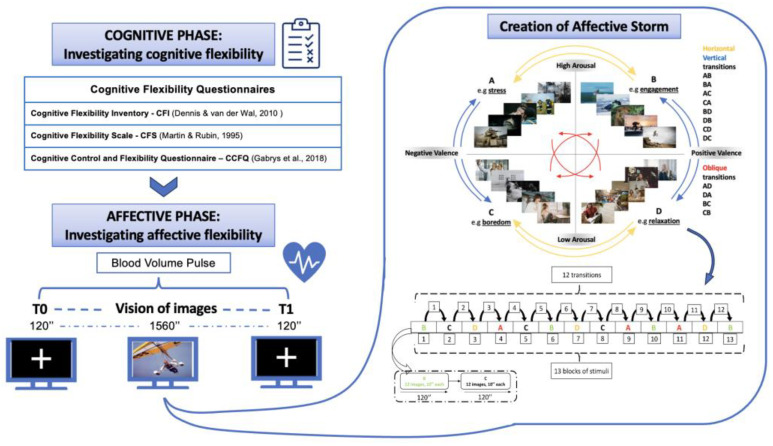
Russell’s [49,50] affect model is enhanced using evocative pictures from the International Affective Picture System (IAPS). A total of 12 potential transitions can be examined between different affect states, inducing an *affective storm*. The transitions include diagonal movements (AD-BC-DA-CB) indicated by red arrows, horizontal movements (AB-CD-BA-DC) indicated by yellow arrows, and vertical movements (AC-BD-CA-DB) by blue arrows. Adapted from [20,21,22,49,50].

**Figure 3 sensors-24-08047-f003:**
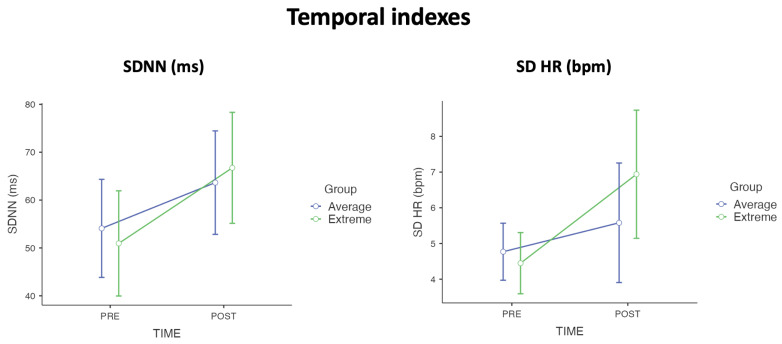
The main effect of time on temporal emotional HRV indexes. Both groups demonstrated in short period indexes (SD HR and SDNN) an increase in variability after the *affective storm* stimulation (points represent the mean and the bars represent the standard deviation).

**Figure 4 sensors-24-08047-f004:**
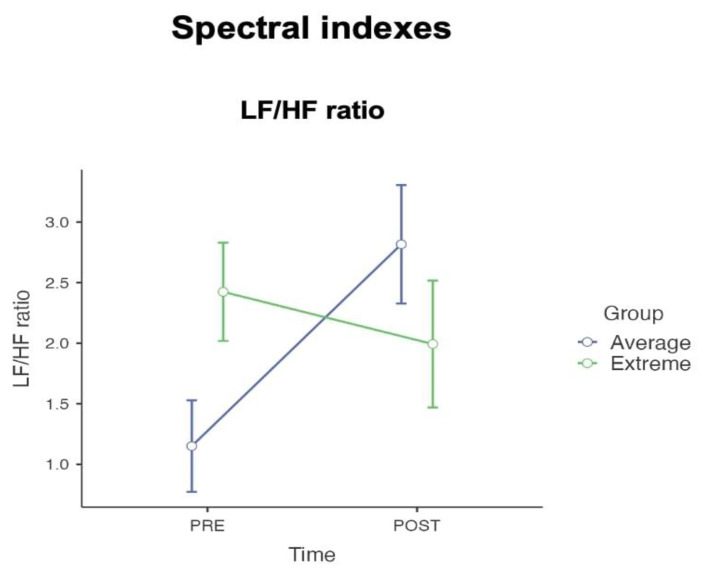
The interaction effect of spectral emotional HRV indexes. Participants with medium levels of cognitive flexibility demonstrated an increase in long-term indexes (e.g., LF/HF) after the *affective storm* stimulation vs. participants with extremely high and low levels of cognitive flexibility showed a decrease in autonomic modulation after the *affective storm* (points represent the mean and the bars represent the standard deviation).

**Figure 5 sensors-24-08047-f005:**
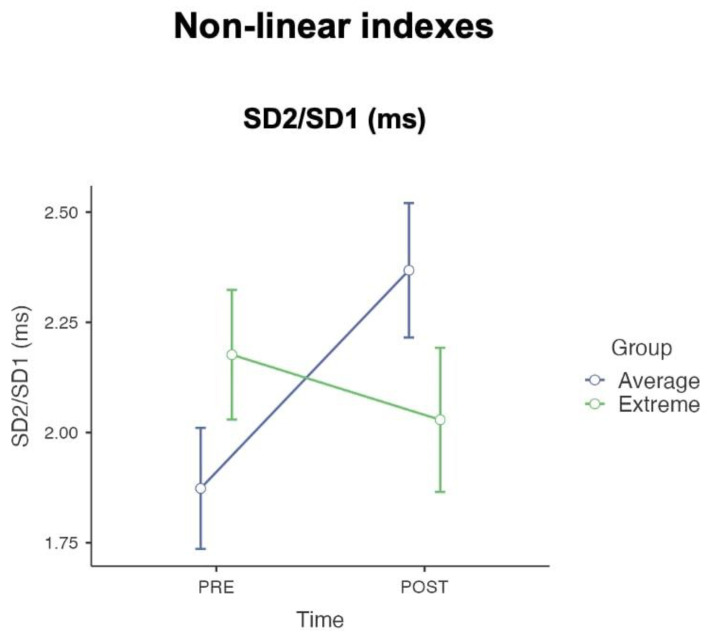
The interaction effect of non-linear emotional HRV indexes. Participants with average levels of cognitive flexibility demonstrated an increase in the long-term/short-term ratio (e.g., SD2/SD1) after the *affective storm* stimulation vs. participants with extremely high and low levels of cognitive flexibility showed a decrease after the *affective storm* (points represent the mean and the bars represent the standard deviation).

**Table 1 sensors-24-08047-t001:** Overview of cognitive flexibility questionnaires.

Questionnaire	Aim	Examples of Items
Cognitive Flexibility Inventory (CFI) [20]	To assess an individual’s cognitive flexibility, specifically their ability to consider alternative perspectives and manage challenges flexibly.	- I am good at putting myself in others’ shoes. - I consider multiple options before making a decision.
Cognitive Flexibility Scale (CFS) [21]	To measure an individual’s perceived control over intrusive negative thoughts and emotions, as well as their ability to respond flexibly to stress.	- I have the self-confidence necessary to try different ways of behaving. - I am willing to work at creative solutions to problems.
Cognitive Control and Flexibility Questionnaire (CCFQ) [22]	To measure cognitive flexibility in interpersonal communication. This scale assesses the adaptability of an individual’s communication style to fit different social situations and the needs of the interaction.	- Generally, in stressful situations, I can remain in control of my thoughts and emotions. - Generally, in stressful situations I take the time to see things from different perspectives before reacting.

**Table 2 sensors-24-08047-t002:** Description of cognitive flexibility variables.

Constructs	Type of Measurements	Measures	Group	N	Mean	Std. Dev.	Std. Error
**Cognitive Flexibility**	**Self-report**	CFI Alternative	Average	24	5.40	0.1132	0.555
			Extreme	20	5.38	0.1919	0.858
		CFI Control	Average	24	4.93	0.1111	0.544
			Extreme	20	4.74	0.2943	1.316
		CFS	Average	24	4.47	0.0903	0.442
			Extreme	20	4.50	0.1605	0.718
		CCFQ Cognitive Control over Emotion	Average	24	35.63	1.0033	4.915
			Extreme	20	35.90	2.5214	11.276
		CCFQ Appraisal and Coping Flexibility	Average	24	45.46	1.1500	5.634
			Extreme	20	45.35	2.4401	10.912

**Table 3 sensors-24-08047-t003:** Description of affective flexibility variables of pre- and post-*affective storm* stimulation.

				T0				T1			
Physio	Type	Indexes	Group	N	Mean	SE	SD	N	Mean	SE	SD
**Heart Rate Variability**	**Temporal Domain**	RMSSD (ms)	Average	23	55.11	7.31	35.07	23	56.13	7.41	35.55
			Extreme	20	44.23	4.20	18.79	20	61.34	5.71	25.52
		SDNN (ms)	Average	23	54.09	5.76	27.64	23	63.64	5.89	28.26
			Extreme	20	50.95	4.43	19.81	20	66.73	4.99	22.30
		SD HR (bpm)	Average	23	4.77	0.43	2.04	23	5.58	0.52	2.48
			Extreme	20	4.45	0.38	1.72	20	6.94	1.16	5.19
		LF power AR (n.u.)	Average	23	47.2	3.61	17.3	23	65.1	4.13	19.8
	**Frequency Domain**		Extreme	20	58.8	4.52	20.2	20	52.5	4.53	20.3
		HF power AR (n.u.)	Average	23	52.8	3.61	17.3	23	34.9	4.12	19.8
			Extreme	20	41.2	4.52	20.2	20	47.4	4.53	20.2
		LF/HF ratio	Average	23	1.15	0.18	0.88	23	2.82	0.40	1.91
			Extreme	20	2.42	0.56	2.49	20	1.99	0.62	2.76
	**Non-linear**	SD1 (ms)	Average	23	39.1	5.20	24.9	23	39.9	5.29	25.4
			Extreme	20	31.4	2.99	13.4	20	43.6	4.05	18.1
		SD2 (ms)	Average	23	65.3	6.56	31.5	23	80.0	6.94	33.3
			Extreme	20	64.2	5.96	26.6	20	83.2	6.22	27.8
		SD2/SD1 ratio	Average	23	1.87	0.12	0.57	23	2.37	0.17	0.81
			Extreme	20	2.18	0.17	0.75	20	2.03	0.14	0.63

## Data Availability

The datasets used and/or analyzed during the current study are available from the corresponding author upon reasonable request.

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
