# Peer review of "In Medio Stat Virtus: Moderate Cognitive Flexibility as a Key to Affective Flexibility Responses in Long-Term HRV"

_sensors, 2024, doi:10.3390/s24248047_

Round 1

Reviewer 1 Report

Comments and Suggestions for Authors

The paper investigates the connection between cogniyive and affective flexibility using both psychometric and physiological measures. It is very interesting and well-written.

I suggest some modification that, in my opinion, can improve the paper.

- In the intro section, I think it is useful to explain in more detail why it is important to study the link between affective and cognitive flexibility.

- In the conclusion section, it may be useful to widen the scope and underline what the results can say at a general level.  

- "According to the DSM-5, both high levels of flexibility—often manifesting as impulsivity—and low levels—manifesting as rigidity—can be dysfunctional, as seen in various clinical conditions such as Attention-Deficit/Hyperactivity Disorder (ADHD), autism spectrum disorders, and eating disorders". This sentence is very interesting and it would benefit of a deeper discussion in the passage between functional and disfunctional behaviours.

- Some details on scales can be useful when they are firstly introduced (the table is very effective in this respect)

- As HRV is central in this study, a dedicated paragraph would help readibility

- "classical null hypothesis significance testing (NHST)", maybe traditional or widely used? please add a reference 

Author Response

Dear Reviewer,

First and foremost, I would like to express our gratitude for the time and effort invested in reviewing my manuscript. Your insightful comments and suggestions have been invaluable in enhancing the quality and clarity of this work. We appreciate the constructive feedback provided, which has significantly contributed to the development of a more robust and well-rounded article. We have carefully considered and addressed each point raised in the revision process.

Thank you once again for your valuable input and guidance.

We highlighted the revisions in green in both the attached reviewer’s answer file and the main manuscript.

General comment: The paper investigates the connection between cognitive and affective flexibility using both psychometric and physiological measures. It is very interesting and well-written.

I suggest some modification that, in my opinion, can improve the paper.

Reviewer 1: In the intro section, I think it is useful to explain in more detail why it is important to study the link between affective and cognitive flexibility.

Author: Thank you for the insightful suggestion to elaborate on the importance of studying the link between cognitive and affective flexibility. We added theoretical and practical implications.

Lines 40-43:Comprehending the interaction of these processes is essential for progressing theoretical frameworks in psychology, since it may uncover common or unique mechanisms that underlie individual adaptation”.
Lines 65-68: “This evidence possesses considerable clinical significance, since deficiencies in cognitive and affective flexibility are characteristic features of different psychopathological conditions. Examining their interconnection may enhance diagnosis accuracy and guide interventions designed to promote adaptation in these domains.”

Lines 129-130: “The inconsistencies emphasized the need for innovative and integrated methodologies that go beyond the separation of cognitive and affective components.”

Reviewer 1: In the conclusion section, it may be useful to widen the scope and underline what the results can say at a general level. 

Author: Thank you for the helpful suggestion! We added a wider potential implication in lines 591-595: The research could potentially contribute to a broader understanding of human adaptability, demonstrating that Flexibility is not merely a cognitive process but a complex interplay of physiological, emotional, and behavioral mechanisms. Analyzing cardiac variability and emotional reactions may reveal the intricate methods by which people adjust their internal states in reaction to external stimuli.”

Reviewer 1: "According to the DSM-5, both high levels of flexibility—often manifesting as impulsivity—and low levels—manifesting as rigidity—can be dysfunctional, as seen in various clinical conditions such as Attention-Deficit/Hyperactivity Disorder (ADHD), autism spectrum disorders, and eating disorders". This sentence is very interesting and it would benefit of a deeper discussion in the passage between functional and disfunctional behaviours.

Author: Thank you for the valuable suggestion!

We added a deeper discussion in Lines 46-60:This relationship between flexibility and functionality follows an inverted U-shaped curve, where optimal functioning occurs at moderate levels of flexibility. At one extreme, excessive flexibility manifests as impulsivity and difficulty maintaining stable patterns of behavior, as seen in ADHD where individuals struggle to inhibit responses and maintain consistent routines (7). This heightened flexibility can lead to difficulties in maintaining focus, completing tasks, and establishing stable behavioral patterns. At the other extreme, insufficient flexibility presents as cognitive and behavioral rigidity, characteristic of autism spectrum disorders, where individuals may struggle to adapt to changes in routine or shift between different perspectives (8). In eating disorders, both extremes can be observed: rigid adherence to strict dietary rules and inflexible thought patterns about body image, or impulsive eating behaviors and difficulty maintaining consistent dietary patterns (10,11). The optimal range of flexibility lies between these extremes, allowing individuals to maintain stable patterns while adapting appropriately to environmental demands.”

Reviewer 1: Some details on scales can be useful when they are firstly introduced (the table is very effective in this respect)

Author: Thank you! We appreciate your observation. We added brief presentations of the scales, trying to avoid repetitions between the introduction and the method sections.

Lines 107-108: “CFI, a 19-item measure divided into 'Alternatives' and 'Control' subscales”.

Lines 112-113: “CFS, a 12-item measure focused on interpersonal communication competence”.

Lines 116-117: “CCFQ an 18-item measure, evaluates both cognitive control over emotions and appraisal/coping flexibility.”

Reviewer 1: As HRV is central in this study, a dedicated paragraph would help readibility

Author: Thank you for your valuable suggestion! We have presented the HRV in the introduction section (line 145 and subsequent). We dedicated a specific paragraph to HRV in the method’s section with the title “Psychophysiological Signal Processing” (line 317).

Reviewer 1: "classical null hypothesis significance testing (NHST)", maybe traditional or widely used? please add a reference

Author: Thank you for your suggestion regarding the terminology and the need for a reference when discussing NHST. We have replaced "classical" with "traditional" to ensure clarity and precision and we added a reference (line 406).

Reviewer 2 Report

Comments and Suggestions for Authors

This empirical study aims to investigate the relationship between cognitive flexibility (CF) and affective flexibility (AF) using a novel method to define the latter. In the literature, the authors stated that AF can be defined using either the executive function framework or the emotion regulation framework. With the executive framework, AF is often measured using the affective task-switching paradigms. From the emotion regulation perspective, the authors mentioned that there are generally two methods to investigate AF: with affective stimuli exposure in laboratory settings or affective changes in naturalistic settings using ESM. The main research gap: mixed results of the AF-CF association, with AF predominantly studied using the executive framework. This research attempted to fill this gap by operationalizing AF differently, from the emotion regulation perspective – as the ability to shift between different emotional states.

I have a few questions at this stage.

Ø  Was the affective storm task developed by the authors? Has this task been validated as an AF task elsewhere? If not, how do we know that the task could successfully induce emotional shifts in participants?

Ø  Relatedly, what is considered an adaptive level of AF or ability to navigate emotional shifts in this study? Did the authors use any behavioral measures in addition to the changes in HRV levels? If it’s just the changes in HRV, would an enhancement (increase in HRV from baseline) or suppression (decrease in HRV from baseline) be considered adaptive?

Ø  In line 274, the authors stated that BVP or PPG signals were used as a metric to assess AF. Typically, HRV reactivity (changes in HRV from baseline to HRV during tasks) is used to investigate the physiological changes as a response to stressors. This was also mentioned in line 144 when the authors discussed findings from another study that examined HRV reactivity and recovery (differences between HRV during tasks and HRV after tasks) with AF. Why did the authors decide to compare pre- and post-experiment HRV instead? Are there prior works that support this quantification of HRV changes?

Ø  One of the main findings was that “individuals with extreme high or low Cognitive Flexibility displayed similar HRV patterns in response to emotional stimuli, while those with average levels of Cognitive Flexibility showed distinct responses”

o   The first part of this finding (extreme groups displayed similar HRV patterns) was examined with group differences in HRV in the pre-experimental condition. The second part of this finding (the average group showed a distinct response) was examined with group differences in HRV changes between pre- and post-experimental conditions.

o   Additionally, the choice of test statistics was different for these two comparisons. The first analysis relied on the Bayes Factor (Bayes t-test) to evaluate the group differences. The second analysis relied on the p-value (“frequentist” ANOVA) to evaluate the group differences

o   Could the authors clarify this inconsistency in the methodology?

Additionally, I also have other more specific comments/questions for the authors:

HRV

Ø  What was the baseline or resting-state protocol used in this study (T0 and T1)? For instance, were the participants instructed to breathe spontaneously? Were they engaging in any vocalization or movement during these phases?

Line 316: SDNN reflects both long-term and short-term variations in heart rate and it is associated with both general emotional states and long-term mood system”

Ø  Both short-term and long-term variations, given a sufficiently long recording. 2-minute is a very short duration. SDNN is likely to predominantly consist of short-term variations

Line 322: We used the LF/HF ratio to determine the balance between sympathetic nervous system (SNS) and parasympathetic nervous system (PNS)

Ø  This is no longer accurate. Please find in-depth discussion in Billman (2013) doi: 10.3389/fphys.2013.00026. For more comprehensive results, authors should consider reporting the associations with HF and LF in addition to other indices – even though 2 minutes is a “bare minimum” to reliably quantify LF. I recommend mentioning that in the manuscript.

Line 471: An increase in SD HR indicates that the heart rate substantially varies during the monitoring period, which is a sign of good heart reactivity and adaptability to external stimuli.

Ø  As mentioned above, reactivity is typically calculated as differences between pre-task and during-task HRV, not with post-task. Regardless, HRV enhancement is not necessarily an indicator of good adaptability to external stimuli. In fact, many studies have proposed the opposite. See Beauchaine et al. (2019) for more details. doi: 10.1111/psyp.13329

Ø  Is there a reason for authors to compute the ratio of SD2/SD1 instead of the conventional SD1/SD2 index?

Presentation of results

Line 362: Afterward, in order to confirm our prior hypothesis, we investigated... The Bayesian t-test results showed that the evidence was greater than 2 for all physiological indicators of Affective Flexibility. This is strong evidence supporting ...

Ø  Should this section be in the Results instead of Methods, especially when it is one of the main findings?

Ø  Authors should standardize the presentation of results across figures. Figures 4 and 5 show both the means and dispersions (is it SD or SE? – authors could clarify in the caption) of HRV indices, while Figure 3 only shows the mean values.

General comments

Ø  The manuscript would benefit from thorough proofreading to improve the consistency in writing. For instance, terms like “average” or “extreme” were capitalized in some instances and not in others. After the first instances of introducing the abbreviations for technical terms like Heart Rate Variability (HRV) or other terms like RMSSD, SNDD etc – the full terms were used and the abbreviations were introduced again on multiple occasions.

Comments on the Quality of English Language

 The English could be improved to more clearly express the research.

Author Response

Dear Reviewer,

First and foremost, I would like to express our gratitude for the time and effort invested in reviewing my manuscript. Your insightful comments and suggestions have been invaluable in enhancing the quality and clarity of this work. We appreciate the constructive feedback, which has significantly contributed to developing a more robust and well-rounded article. We have carefully considered and addressed each point raised in the revision process.

Thank you once again for your valuable input and guidance.

We highlighted the revisions in green in both the attached reviewer’s answer file and the main manuscript.

General comment review: This empirical study aims to investigate the relationship between cognitive flexibility (CF) and affective flexibility (AF) using a novel method to define the latter. In the literature, the authors stated that AF can be defined using either the executive function framework or the emotion regulation framework. With the executive framework, AF is often measured using the affective task-switching paradigms. From the emotion regulation perspective, the authors mentioned that there are generally two methods to investigate AF: with affective stimuli exposure in laboratory settings or affective changes in naturalistic settings using ESM. The main research gap: mixed results of the AF-CF association, with AF predominantly studied using the executive framework. This research attempted to fill this gap by operationalizing AF differently, from the emotion regulation perspective – as the ability to shift between different emotional states.

I have a few questions at this stage.

Reviewer 2: Was the affective storm task developed by the authors? Has this task been validated as an AF task elsewhere? If not, how do we know that the task could successfully induce emotional shifts in participants?

Authors: We sincerely appreciate your insightful observations regarding the methodological context of our affective storm task. We are grateful for the opportunity to provide additional clarity on the task's development and theoretical foundations.

Indeed, the affective storm task was originally developed by our research team and was first presented in a protocol article by Borghesi et al. (2023), subsequently employed in an experimental design by Simoncini et al. (2024). In conceptualizing the task's implementation, we drew significant inspiration from the sophisticated experimental paradigms developed by Genet, Malooly, and Siemer (2011, 2012, 2013), who pioneered the use of emotional stimuli from the International Affective Picture System (IAPS) database to investigate affective processes.

While Genet and colleagues operationalized affective flexibility primarily through switching cost metrics, we have thoughtfully recontextualized their foundational approach. Our adaptation aligns more closely with the theoretical definition proposed by Hollenstein (2013) and Koval (2015), specifically conceptualizing affective flexibility as the ability to transition between different affective states.

We added these clarifications in lines 278-282: “This task, previously developed (Borghesi et al., 2023) and applied to an experimental study (Simoncini et al., 2024), relies on the idea that Affective Flexibility can be examined through the lens of affective dynamics, reflecting the notions posited by Hollenstein (2013) and Koval (2015), who conceptualized Affective Flexibility as the capacity to shift between different affective states”.

Reviewer 2: Did the authors use any behavioral measures in addition to the changes in HRV levels?

Authors: Thank you for the valuable question! We intentionally did not interrupt participants' emotional experience during stimulus presentation to preserve the ecological validity of the emotional response. Existing literature suggests cardiac variability positively correlates with emotional adaptability and psychological well-being (Weber et al., 2010; Colzato et al., 2018; Zahn et al., 2016). We added a specific part in the discussion section in lines 508-520: “This increase could reflect short-term cardiac physiological immediate response in terms of recovery following exposure to the continuous stimulation of an affective storm. These results align with previous studies on emotions, which have similarly reported heightened activations during and post-task phase under comparable conditions (70,71) Heightened activations may serve as a generalized adaptive mechanism, reflecting a short-term cardiac physiological response aimed at recovery following the continuous stimulation of an affective storm. Such short-term cardiac responses likely represent a broader pattern of recovery dynamics, where the physiological system momentarily amplifies activity to address the challenges presented by extreme emotional stimuli (72,73). The experimental design, which included highly evocative stimuli varying in emotional intensity, supports this interpretation, as it highlights the role of extreme stimuli in eliciting pronounced, albeit temporary, physiological adjustment.”

However, the spectral indexes present a more intricate picture of autonomic response that can be meaningfully interpreted through the lens of well-being. Given that we utilized selected International Affective Picture System (IAPS) images with predetermined emotional intensity scores, we anticipated differential physiological responses. Specifically, we hypothesized that participants with moderate emotional flexibility would exhibit an increased autonomic ratio following exposure to affectively challenging stimuli. Conversely, participants with high initial flexibility—characterized by an already elevated sympathetic-vagal ratio—might demonstrate a subtle downregulation of this response. This pattern suggests that the emotional stimulation may not have been sufficiently intense to disrupt their baseline physiological state substantially.

Reviewer: Relatedly, what is considered an adaptive level of AF or ability to navigate emotional shifts in this study?  If it’s just the changes in HRV, would an enhancement (increase in HRV from baseline) or suppression (decrease in HRV from baseline) be considered adaptive?

Authors: Thank you for your valuable observations! The question of what constitutes adaptive levels of autonomic flexibility (AF) or emotional shift navigation in this study requires careful consideration. The relationship between HRV changes and adaptivity is complex and context-dependent. When examining HRV indexes, we observe an intriguing pattern: the extreme group shows decreased variability in long-term index, while the average group displays an opposite trend. This distinction is particularly noteworthy as it represents the first study to analyze spectral long-run index in this context, making our findings novel and challenging to contextualize within the existing literature. It's crucial to differentiate between adaptivity and functionality. The decreased variability observed in the extreme group might actually represent an adaptive mechanism - a strategic autonomic response - rather than a deficit. However, to fully understand the functional implications of these patterns, we need to examine their manifestation in clinical populations and their relationship to real-world outcomes. The fact that we see these contrasting patterns specifically in long-term indexes is particularly significant. These opposing tendencies suggest distinct autonomic regulation strategies between groups, though their relative advantages or disadvantages may depend on environmental demands and individual circumstances.
Our study is the first to look at spectral long-term indexes, so we don’t have other studies to compare.

We added these considerations in the Discussion section in lines 532-538: “This distinct psychophysiological behavior between the Average and Extreme groups recalls broader principles of human adaptation and functionality. The former pertains to the individual's adaptive reactions to environmental situations, whilst the latter underscores the significance of these responses in improving the individual's well-being in a specific context. Thus, adaptability does not inherently reflect a functional response to a particular setting. In this context, both physiological behaviors observed in the spectral and nonlinear indices of the two groups can be considered adaptive. Extreme individuals exhibited higher baseline sympathetic activation, which, following the affective storm, could only remain constant or decrease. In contrast, Average individuals started from lower levels of sympathetic activity, affording them greater degrees of freedom to increase their activation after the storm. Both patterns appear adaptive, as they reflect appropriate responses to the stimuli and the respective baseline states of the groups. The key question, however, lies in determining which of these responses is also functional. Future research, particularly involving clinical populations, will be critical in exploring this aspect and understanding how these adaptive patterns relate to well-being in specific contexts.

And in the conclusion section in lines 580-583: “These disparities indicate opposing adaptation processes adopted by the two groups, the actual functionality of which remains unknown. Future research could include clinical populations to examine the dysfunctional nature of some adaptations to environmental stimuli.”

Reviewer 2: In line 274, the authors stated that BVP or PPG signals were used as a metric to assess AF. Typically, HRV reactivity (changes in HRV from baseline to HRV during tasks) is used to investigate the physiological changes as a response to stressors. This was also mentioned in line 144 when the authors discussed findings from another study that examined HRV reactivity and recovery (differences between HRV during tasks and HRV after tasks) with AF. Why did the authors decide to compare pre- and post-experiment HRV instead? Are there prior works that support this quantification of HRV changes?

Authors: Thank you for pointing out this fundamental characteristic of our methodology. Our experimental design investigates the effects of the affective storm, a protocol where participants experience rapid transitions between emotional extremes (valence and arousal). When measuring responsiveness during this protocol, we must account for self-canceling blocks that occur as participants transition between positive and negative states and between high and low arousal ones. These alternating affective states create oscillations that influence cardiac variability. Furthermore, our design has a notable temporal asymmetry: the affective storm exposure phase lasts 26 minutes, while each baseline phase is 2 minutes long. This substantial difference in duration may affect cardiac indexes and could potentially complicate direct comparisons between phases. We adhered to the standard structure commonly seen in Affective Flexibility studies. For instance, Grol and De Raedt (2020) examined HRV exclusively in the resting state, while their subsequent study (2021) explored HRV across pre-, during-, and post-stimulus phases. However, the affective stimulus in the latter study was unidirectional, focusing solely on stress-inducing conditions. Additionally, previous studies by Souza et al., (2007) and Sokhadze (2007) implemented a pre-, during and post- stimulation design in order to explore recovery functions of HRV.

Reviewer 2: One of the main findings was that “individuals with extreme high or low Cognitive Flexibility displayed similar HRV patterns in response to emotional stimuli, while those with average levels of Cognitive Flexibility showed distinct responses”

The first part of this finding (extreme groups displayed similar HRV patterns) was examined with group differences in HRV in the pre-experimental condition. The second part of this finding (the average group showed a distinct response) was examined with group differences in HRV changes between pre- and post-experimental conditions.

Additionally, the choice of test statistics was different for these two comparisons. The first analysis relied on the Bayes Factor (Bayes t-test) to evaluate the group differences. The second analysis relied on the p-value (“frequentist” ANOVA) to evaluate the group differences

 Could the authors clarify this inconsistency in the methodology?

Authors: We appreciate your observation regarding the methodological differences in our analyses. We acknowledge that the primary outcome of our study focuses on the differences between the Average and Extreme groups in terms of HRV responses. However, before grouping the Extreme High and Extreme Low participants into a single Extreme group, it was essential to demonstrate their physiological similarity to ensure the validity of this aggregation.

To address this, we employed a Bayesian t-test to evaluate the similarity in physiological responses between the Extreme High and Extreme Low subgroups in the pre-experimental condition. We chose the Bayesian approach for two reasons:

  1. Sampling Considerations: The Extreme High and Extreme Low subgroups were derived from a sample of 44 participants, each consisting of 10 individuals. Given the relatively small size of these subgroups, we considered Bayesian statistics to be more appropriate, as they do not require assumptions of normality in the population. This choice provided a robust method for assessing similarity without over-relying on parametric assumptions.
  2. Focus on Similarity: The Bayesian Factor allowed us to effectively quantify evidence for similarity between the Extreme subgroups, which was the critical objective of this analysis. By confirming their similarity, we could justify combining these subgroups into a single Extreme group for subsequent analyses.

We used the frequentist ANOVA to assess the differences between the Average and Extreme groups in pre- and post-experimental conditions because it is a well-established method for analyzing variance across conditions, particularly when the sample size and design permit.

To streamline the results section and avoid overburdening the reader, we opted to present the Bayesian analysis as a confirmatory step rather than a primary result. However, all Bayesian analyses, along with supplementary data, have been included in the supplementary materials for transparency and reproducibility.

We hope this clarifies the rationale behind our methodological choices. Please let us know if further clarification or elaboration is needed.

Reviewer 2:  What was the baseline or resting-state protocol used in this study (T0 and T1)? For instance, were the participants instructed to breathe spontaneously? Were they engaging in any vocalization or movement during these phases?

Authors: Thank you for your thoughtful questions regarding our experimental protocol. Regarding the resting state phases, we implemented a protocol where participants were presented with a white fixation cross displayed on a black background via a computer screen. To maintain optimal data quality, participants were instructed to maintain a relaxed state by keeping their gaze on the fixation cross while breathing naturally. We specifically requested that participants remain still and refrain from verbal communication during both resting state periods to minimize motion artifacts and ensure consistent baseline measurements.

We added the information in lines 297-302: “During the resting state phases, participants were presented with a standardized visual stimulus consisting of a white fixation cross displayed centrally on a black background via a computer monitor. Subjects were instructed to maintain their gaze on the fixation point while engaging in spontaneous respiration, with explicit directions to refrain from verbal communication or movements to minimize artifacts.”

Reviewer 2: Line 316: SDNN reflects both long-term and short-term variations in heart rate and it is associated with both general emotional states and long-term mood system”

Both short-term and long-term variations, given a sufficiently long recording. 2-minute is a very short duration. SDNN is likely to predominantly consist of short-term variations.

Authors: Thank you for pointing out this important observation. You are absolutely correct that SDNN, given the 2-minute duration of our recording, primarily reflects short-term variations rather than both short- and long-term variations. This was an oversight, and we have corrected the terminology in the manuscript to reflect this appropriately.

Additionally, we have included a note in the limitations section acknowledging the 2-minute duration as a potential limitation of our study. Practical considerations drove our choice of this time frame: we aimed to maintain a balance between ensuring sufficient data collection for HRV analysis while avoiding participant fatigue, particularly given the emotionally evocative stimuli presented before and after the baseline recordings.

We inserted this part in the Limitations section, in lines 608-613:  “An additional limitation concerns the 2-minute duration of our HRV resting-state recordings. While this timeframe allowed us to minimize participant fatigue during exposure to emotionally evocative stimuli, longer recording periods might have provided more robust HRV data. Future studies may benefit from exploring extended measurement intervals while carefully considering the trade-off between comprehensive physiological assessment and participant engagement.”

Reviewer 2: Line 322: We used the LF/HF ratio to determine the balance between sympathetic nervous system (SNS) and parasympathetic nervous system (PNS)

This is no longer accurate. Please find in-depth discussion in Billman (2013) doi: 10.3389/fphys.2013.00026. For more comprehensive results, authors should consider reporting the associations with HF and LF in addition to other indexes – even though 2 minutes is a “bare minimum” to reliably quantify LF. I recommend mentioning that in the manuscript.

Authors: Thank you for raising this critical point and referencing the comprehensive discussion by Billman (2013). We recognize the limitations of the LF/HF ratio as a measure of cardiac sympatho-vagal balance, particularly given its simplifying assumptions and the increasing body of literature questioning its reliability.

In addition to incorporating the recommendations from Billman (2013), we have also reviewed the work of von Rosenberg et al. (2017), which further elaborates on the limitations of univariate metrics like LF/HF and introduces a two-dimensional representation of LF and HF bands.

In our revised manuscript, we have:

  1. Clarified that the LF/HF ratio was used as a general indicator of autonomic modulation rather than a precise measure of sympatho-vagal balance, in method, results, discussion and conclusion sections.
  2. Acknowledged the interpretative challenges of the LF/HF ratio, as highlighted in the literature, including its limited capacity to fully capture autonomic dynamics. Lines 354-356: ​​”This index implicates interpretative challenges, since it shows limited capacity to fully capture autonomic dynamism in terms of balance between sympathetic and parasympathetic system.
  3. Cited von Rosenberg et al. (2017) to propose future research directions that might incorporate advanced methodologies, such as 2D scatter plots, to overcome these limitations.

Furthermore the observation, noted by both Billman and von Rosenberg—"a high SDNN can be caused by an increased activity of either the SNS, the PNS, or both"—is absolutely correct and aligns perfectly with our findings. Indeed, in terms of short-term variability, SDNN increases in both groups. However, when examining autonomic behavior, the patterns diverge: in the Extreme group, autonomic activity either decreases or remains relatively stable, whereas in the Average group, it increases. This divergence suggests that the key factor driving these differences in autonomic response lies in group membership. The physiological impact of the affective storm appears to interact with the distinct baseline and adaptive characteristics of each group, leading to differing autonomic effects. This reinforces the importance of considering both group dynamics and autonomic influences when interpreting the physiological outcomes of emotional challenges.

Regarding the 2-minute duration, we agree with the reviewer that this is a “bare minimum” for reliable LF quantification. This constraint was necessitated by the practical requirements of our study design, as longer recordings risked participant fatigue and diminished data quality. We have acknowledged this limitation explicitly in the Limitations section, precisely in lines 603-608: “Regarding HRV indexes, while we utilized the LF/HF ratio as one of our metrics, it is important to acknowledge ongoing debates in the field regarding its interpretation as a complex index. Future research might incorporate advanced methodologies, such as 2D scatter plots, to address these limitations and provide a more nuanced understanding of autonomic modulation.” 

Reviewer 2: An increase in SD HR indicates that the heart rate substantially varies during the monitoring period, which is a sign of good heart reactivity and adaptability to external stimuli.

As mentioned above, reactivity is typically calculated as differences between pre-task and during-task HRV, not with post-task. Regardless, HRV enhancement is not necessarily an indicator of good adaptability to external stimuli. In fact, many studies have proposed the opposite. See Beauchaine et al. (2019) for more details. doi: 10.1111/psyp.13329

Authors: Thank you for highlighting these critical points. We completely agree that the current phrasing regarding heart rate reactivity and adaptability is not consistent with standard definitions, which typically calculate reactivity based on differences between pre-task and during-task HRV, rather than post-task measures. We have therefore removed the reference to "reactivity" in this context to avoid any potential misinterpretations.

Moreover, you are absolutely correct that HRV enhancement is not universally an indicator of good adaptability to external stimuli, and in some cases, it may indicate the opposite, as highlighted in Beauchaine et al. (2019). This insight has prompted us to critically reevaluate some of our assumptions, particularly regarding the distinctions between Average and Extreme groups regarding adaptability and functionality. To better address these distinctions, we have refined our discussion and incorporated additional theoretical perspectives, including the insightful arguments presented by Schmitt and Pincher (2004) the framework provided in Evaluating Evidence of Psychological Adaptation: How Do We Know One When We See One (Schmitt and Pincher, 2004) These sources have helped us delineate more clearly between physiological adaptability and the functional outcomes associated with different HRV patterns.

Lines 532-538: “This distinct psychophysiological behavior between the Average and Extreme group recalls broader principles of human adaptation and functionality. The former pertains to the individual's adaptive reactions to environmental situations, whilst the latter underscores the significance of these responses in improving the individual's well-being in a specific context. Thus, adaptability does not inherently reflect a functional response to a particular setting”

Reviewer 2: Is there a reason for authors to compute the ratio of SD2/SD1 instead of the conventional SD1/SD2 index?

Authors: Thank you for your comment and for pointing out the convention of using the SD1/SD2 index instead of the SD2/SD1 ratio. Our decision to use the SD2/SD1 ratio was intentional and grounded in the specific goals of our study. The SD2/SD1 ratio emphasizes the relationship between long-term (SD2) and short-term (SD1) variability of heart rate intervals. This ratio is particularly informative when assessing the dominance of autonomic components over longer time scales relative to instantaneous fluctuations. By focusing on SD2/SD1, we aimed to align our analysis with the study's theoretical framework, which emphasizes longitudinal adaptability rather than acute reactivity. This approach allows for a more nuanced interpretation of the balance between parasympathetic and sympathetic modulation over extended emotional transitions.

Furthermore, the SD2/SD1 ratio has been referenced in the scientific literature (e.g., Bodenes et al., 2022; Balocchi et al., 2006) as a measure capable of capturing patterns indicative of systemic autonomic regulation. In this context, the choice of SD2/SD1 reflects a deliberate focus on the patterns most relevant to our hypotheses.

Presentation of results

Reviewer 2: Line 362: Afterward, in order to confirm our prior hypothesis, we investigated... The Bayesian t-test results showed that the evidence was greater than 2 for all physiological indicators of Affective Flexibility. This is strong evidence supporting ...

Should this section be in the Results instead of Methods, especially when it is one of the main findings?

Authors: Thank you for your thoughtful comment. We understand your concern about the placement of the Bayesian t-test results, as these analyses do play an important role in supporting our methodology. However, we opted to include this analysis in the Methods section because it serves primarily as a confirmatory step to validate the physiological similarity between the Extreme High and Extreme Low groups, allowing us to aggregate them into a single Extreme group. This step was essential to our study design but was not intended as a primary result. To clarify this choice, we have revised the Methods section to explicitly state that the Bayesian t-test was used as a confirmatory analysis to support the grouping methodology. Additionally, we have included the full results of these analyses in the supplementary materials for transparency, ensuring that readers can access the details without disrupting the flow of the Results section.

Our intention was to avoid overcomplicating the narrative of the Results by including intermediate findings that are methodological in nature rather than central to our primary outcomes. We hope this explanation justifies our approach, but we are happy to make further adjustments if needed.

Reviewer 2: Authors should standardize the presentation of results across figures. Figures 4 and 5 show both the means and dispersions (is it SD or SE? – authors could clarify in the caption) of HRV indexes, while Figure 3 only shows the mean values.

Authors: Thank you for bringing this to our attention! Figure 4 and 5 show means and SD of HRV indexes. We added them also to Figure 3 and we specified SD.

Reviewer 2:

The manuscript would benefit from thorough proofreading to improve the consistency in writing. For instance, terms like “average” or “extreme” were capitalized in some instances and not in others. After the first instances of introducing the abbreviations for technical terms like Heart Rate Variability (HRV) or other terms like RMSSD, SNDD etc – the full terms were used and the abbreviations were introduced again on multiple occasions.

Authors: Thank you for bringing this to our attention! We have changed the formatting regarding abbreviations within the text so that they are introduced only the first time, to avoid repetition. Additionally, in order to standardize the formatting of the terms “Average” and “Extreme,” under his advice, we have retained capital letters when we specifically refer to the two groups, while instead opting to use synonyms (e.g., “medium” instead of “average”) in other cases.

Round 2

Reviewer 2 Report

Comments and Suggestions for Authors

I appreciate the authors taking the time to address the comments and suggestions. I have no further comments on the manuscript.

Author Response

Thank you for your revisions! Your feedback has been instrumental in helping us enhance the manuscript, making it more thorough and balanced.